# Needs of Older People Attending Day Care Centres in Poland

**DOI:** 10.3390/healthcare8030310

**Published:** 2020-08-29

**Authors:** Justyna Mazurek, Dorota Szcześniak, Elżbieta Trypka, Katarzyna Małgorzata Lion, Renata Wallner, Joanna Rymaszewska

**Affiliations:** 1Department and Division of Medical Rehabilitation, Wroclaw Medical University, 50-367 Wroclaw, Poland; 2Department of Psychiatry, Wroclaw Medical University, 50-367 Wroclaw, Poland; dorota.szczesniak@umed.wroc.pl (D.S.); elzbieta.trypka@umed.wroc.pl (E.T.); k.urbanska@hotmail.com (K.M.L.); renata.wallner@umed.wroc.pl (R.W.); joanna.rymaszewska@umed.wroc.pl (J.R.); 3Menzies Health Institute Queensland, Griffith University, Brisbane 4111, Australia

**Keywords:** elderly, health and social care, community, day care centres, health promotion

## Abstract

Introduction: Day care centres (DCC) aim to support older adults living in their own homes by providing a variety of activities to promote the independence of those people and reduce the caregiver’s burden. In Poland, there are no standards for providing this form of care. The provided care is delivered by different organisations, and there is a lack of quality control in the majority of places. Regrettably, in Poland, there is a paucity of research on the holistic needs of the elderly attending DCC. Aim of this study: This is the first study which has aimed to identify the Polish day care centres attendees’ needs to ensure that the increasing number of older people receive the best possible care, and as a part of the quality improvement process for recommendation development of the new day care services model in Poland within the ‘Homely Marina’ project. Methods and Materials: A representative sample (*n* = 269) was randomly selected from day care centres attendees (estimated as 10,688) in Poland. An anonymous survey for the assessment of needs was developed by the authors. Results: The respondents assessed the level of provided services as very good or good. Best rated services included meals, supportive and welfare services (occupational therapy, music therapy, art therapy, cognitive training). Almost half of the participants indicated the need for company as the main reason to attend a DCC. This research reveals a lack of support with regard to additional staff: e.g., a nurse. Conclusions: In Poland, the services offered in DCC should focus on social isolation and sense of loneliness prevention, and maintenance of social activity of the elderly. The presented analysis of needs in Polish day care centres suggests a need for changes which may improve the quality of services. There is a great need to find a balance between home-based care and in-patient care, using better integration of available services and strengthening support for informal caregivers. Robust research with a collection of meaningful outcomes is required to ensure that in Poland, the increasing number of older people is enabled to access high-quality day care service provision.

## 1. Introduction

Currently, there are over 6 million people above 65 years living in Poland, which constitutes about 17.5% of the Polish population. In 2017, the average life expectancy was estimated as 73.96 years for men and 81.82 years for women, and it continues to increase [1], resulting in the perspective of a growing number of older adult population in the next decades. Solitary living of older people, the so-called singularisation of old age [2], is becoming more and more popular and is a significant consequence of demographic ageing of the Polish society.

Predicted changes in the demographic structure will result in several challenges in planning and shaping the social policy related to older people. The ageing population will require more organised help to support daily activities and improvement of general health among elderly people. Therefore, local and state authorities in Poland will need to face these challenges, undertake actions and meet seniors’ broad and complex needs [3].

The most important actions that should be provided in order to meet elderly people’s needs include essentially social activation, including health-promoting activities like physical activity, kinesitherapy, educational, cultural, recreational and welfare offers adjusted to the needs stated in the local environment, transportation solutions for older adults in isolated communities, adequate access to care, good infrastructure and coordination of services, scarce assisted living and in-home care for frail older adults, and minimizing barriers related to culture, language, and economics [4,5,6,7,8].

The World Health Organisation (WHO) proposes a complex definition of health as “a state of complete physical, mental, and social well-being and not merely the absence of disease or infirmity” [9]. They do not define the terms of mental or social well-being in details. However, separate definitions of these terms can be applied. “Mental health” can be understood as a sense of coherence which contributes to the ability to effectively manage the experienced stressful situations and the ability to recover quickly in terms of mental balance when it is in danger [10]. Additionally, “social health”, however, encompasses three areas related to: (1) the ability to realise one’s own potential and fulfil one’s obligations, (2) the ability to manage one’s own life with a certain level of independence (autonomy) regardless of health problems and (3) the ability to participate in social activities [11]. The World Health Organization includes and summarises all the aspects described above within its definition of “active ageing”, which is a “process of optimizing opportunities for health, participation and security in order to enhance the quality of life as people age” [12]. Therefore, it is essential to pay attention to the improvement, not only to the physical health, but also to the mental and social well-being of older people. It can be achieved by strengthening the role of social security.

In Poland, long-term care for elderly people is scattered through various financial and provision systems. The services are delivered by the health care system (Narodowy Fundusz Zdrowia; NFZ), the social care system and additionally by the private sector and non-governmental organisations. The public social care system provides services in (1) in-patient and (2) out-patient settings as well as (3) at individuals’ homes [13].

One of the most popular forms of supporting older people are day care centres (DCC). Day care centres offer a range of activities, including health care services like physical or occupational therapy, usually adjusted to the needs of their attendees, such as a social club and training. Additionally, they can arrange some trips and outdoor activities [14,15,16].

In Poland, DCCs are managed by the local governments in cooperation with social care centres. This form of support is dedicated to community-dwelling people (mostly elderly), who might need minor support in activities of daily living (ADL). The fee for participation in the day care is established by the government. Depending on the participant’s individual (or family) annual income, the fee can be partially or fully covered by the social welfare services. These places usually operate five days a week (no longer than 12 h a day). The day care centres have a great potential to play a significant role in Poland in promoting social health and improving general well-being among older people.

Although in Poland, day care centres function on the basis of the Social Assistance Act [17], detailed principles regarding the functioning of DDCs have not been precisely regulated yet. Each DDC functions on the basis of the resolutions of particular commune councils which constitute the so-called local law, and specify, e.g., particular conditions of granting services and fees for services provided at a given DCC, principles of partial or total exemption from fees and the mode of charging fees. Most frequently, DCC is directly subject to a manager who organises and supervises work while paying attention to safe and hygienic working conditions of all subordinate employees. In Poland, legal provisions do not specify the standards and principles of the functioning of day care centres, including, for example, required infrastructure, qualifications of the personnel, standards regarding the number of attendees per one employee of the day care centre, daily routine, etc. (such information are specified to a various extent in the statutes and rules and regulations of a given centre). Day care centres do not have a formalised way of assessment of their functioning. Therefore, there is a lack of appropriate standards, schedules, frequency of assessment and a possibility of objective verification of the implemented actions as well as the assessment of the needs of attendees of DCCs in Poland and their satisfaction with the undertaken actions.

The studies measuring outcomes for older people attending day care centres are very limited in terms of outcome data. A Canadian study [18] suggested that attendance at a day care centre could reduce hospital attendance and admissions. There are no studies in Poland measuring the outcome data and complex needs of this group of people. Little is known about different models of day care and their impact on the experience of those accessing services and whether certain models of day care may demonstrate greater benefit, and therefore, be more appropriate for the future needs of older people. Although the respite function of day care is acknowledged and explored, less is known about the complex needs of the participants.

Unfortunately, informal long-term care is still the most common form of supporting older people in Poland. It is based on voluntary support provided by family members or friends. It is also the most cost-effective form of care [13,19].

The study objective was to identify the day care centres attendees’ needs in various locations in Poland to ensure that the increasing number of older people receives the best possible care, and as a starting point for recommendation development of the new day care services model within the EU funded ‘Homely Marina’ project (Polish: “Domowa przystań”). The project (2016–2020) aimed to adapt and implement the Italian model (the Emilia Romagna region) of day care centres for older adults into Poland (the Lower Silesia region). The new support model was based on the solutions implemented by Azienda Pubblica Di Servizi Alla Persona Citta Di Bologna; ASP in Italy (foreign project partner) and concerned the expansion and improvement of the quality of interdisciplinary services in the following areas: care, welfare and affliction prevention in older people.

## 2. Materials and Methods

### 2.1. Study Design

This was a quality improvement process which involved collecting day care centres’ participants data regarding the future implementation of the new support model. Based on the policy activities that constitute the research at Wroclaw Medical University, this work met the criteria for operational improvement activities exempt from ethics review. This project was reviewed by the Municipal Social Service Centre in Wroclaw (in Polish: Miejskie Centrum Usług Socjalnych, MCUS) institutional review board (project identification code: POWR.04.03.00-IP.07-00-004/16). However, the investigations were carried out following the rules of the Declaration of Helsinki of 1975, revised in 2013.

The data was collected between September and November 2017.

### 2.2. Survey

An anonymous survey entitled “Needs assessment of day care centres attendees in Poland” (Appendix A) was developed by Wroclaw Medical University researchers in cooperation with the Municipal Social Service Centre in Wroclaw project staff. It consisted of the title page, a section covering sociodemographic data and a needs assessment part. The survey included open, semi-open and closed questions. Its feasibility was piloted among five, randomly chosen, day care centre participants in Wroclaw. There was no need for any changes after piloting the survey. The questionnaire for the purposes of the article has been translated into English and constitutes an attachment to the presented work.

The survey was developed based on the official care guidance for day care centres in Poland [17] as well as the guide for a new support model from Italy. The following categories of suggested services, actions and activities were used:(1)welfare services and specialist welfare services, defined as support in meeting daily life needs and, as long as it is possible, ensuring contacts with the environment. The welfare services and specialist services include, among other things, occupational therapy (including music therapy, art therapy, dance movement therapy, ergotherapy, etc.), cognitive training, support in personal or administrative issues; specialist welfare services should be adjusted to the specific needs related to an affliction or disability, and they are provided by qualified specialists;(2)care and hygienic services are connected with providing access to the equipment improving personal hygiene (i.e., shower, washing machine, etc.); monitoring blood pressure, sugar level and weight; registered nurse’s services (e.g., help in dividing and administering drugs); and care activities like help in maintaining personal hygiene, taking care of appearance and cleanliness at person’s home, help in washing, taking baths and getting dressed;(3)preventive services that encompass meeting the health needs established through the contact with health care providers like nurses, psychologists, physiotherapists that may improve mobility (e.g., through movement classes in a broad sense, kinesiotherapy) as well as educational activities and lectures conducted by invited guests, e.g., medical practitioners (doctors, nurses, physiotherapists) and health care specialists;(4)supportive actions described as facilitating meeting social, cultural, recreational (games, plays) tourism (organising leisure activities and trip), needs, e.g., participation in cultural events;(5)and other ones, that is services provided outside the above-mentioned main categories, i.e., computer classes, foreign language learning, access to the library and current press, ensuring transportation of the attendees from the place of residence to the day care centre and back, classes for informal caretakers (mostly support groups) as well as delivering meals to the place of residence of persons who do not avail themselves of the remaining part of the offer of the day care centre.

### 2.3. Cluster Randomisation

A representative sample of day care centre participants was cluster-randomly selected based on an estimated number of 10,688 attendees in Poland. Assuming the level of confidence of 95%, fraction size of 0.6, a maximum error of 5%, the size of the sample amounted to 356 people. The simple random sampling was applied, consisting of the direct and unlimited sampling of the studied entities for a statistical sample directly from the population and without any limitations.

The randomisation was conducted with the use of a box for drawing lots. Particular studied entities were replaced with lots (numbers) and then placed inside the drawing box. After mixing, the lots were drawn with the preservation of all and any probability rules and the appropriate number of lots needed for the research. The sample selected in such a manner was characterized by all the features of a representative sample. Figure 1 presents the recruitment process of participants in detail.

### 2.4. Participants

As many as 1960 questionnaires were sent by postal service to the chosen (in a randomised way) day care centres in various cities in Poland, i.e., Wrocław, Łódź, Sochaczew, Konin, Białystok and Bytom. The total of 269 questionnaires were returned from the centres located in Wrocław (155 from 6 centres), Łódź (54 from 2 centres) and Bytom (60 from 4 centres). It constituted 14% of all posted surveys.

The participants were recruited from those who used day care centres in Poland. Obtaining data from various locations in Poland was crucial due to the fact that these are places of support carried out by local government units. Thus, covering the survey with people from different cities makes it possible to collect comprehensive data. All the people were informed about the research objective, and their written consent was obtained. The study was conducted once. If the elderly person did not understand the question, a previously trained person from the staff explained the question and doubts.

The inclusion criteria for respondents were based on the formal requirements for participation in day care centres in Poland [17]. The detailed criteria covered people who, based on their age (over 60 years old), disease or disability, demand partial care and help in meeting necessary life needs. The exclusion criteria included the lack of consent to participate in it.

## 3. Results

### 3.1. Characteristics of the Studied Group

The authors prepared features included in Table 1 in this form by conducting a literature review on this topic. The mean age of study participants (*n* = 269) was 74.1 years (±10.6). Twenty four (9%) people did not state their year of birth. The majority (61%; *n* = 164) of the group identified themselves as women. Slightly more than a half (51%; *n* = 136) of all the respondents were widowed, and 12% (*n* = 32) were divorced. The majority of people who participated in this study were living alone.

The majority of the respondents completed secondary (37%; *n* = 100) and vocational education (30%; *n* = 80). Almost one-third of the respondents (*n* = 86; 32%) described themselves as physical workers (blue-collar workers) in the past. The response to the question regarding a previously practised profession was not given by as many as 77 (29%) participants. As many as 176 respondents (65%) were retired (pension), whereas 60 people (22%) received a disability benefit (disability pension). The vast majority (*n* = 181; 67%) of the respondents lived in large cities (over 300,000 inhabitants).

Most respondents (*n* = 129; 48%) had used the services of day care centres for more than 5 years at the moment of completing the survey. The availability of day care centres was assessed as very good (*n* = 99; 37%) and good (*n* = 76; 28%) by 65% of the respondents. Detailed sociodemographic characteristics of the participants are presented in Table 1.

### 3.2. Needs within the Scope of the Services of Day Care Centres

#### 3.2.1. Assessment of Provided Services among All Study Participants

Within all the respondents (*n* = 269), the three best assessed particular services were: meals (56%; *n* = 151), supportive services (social gatherings, socializing events, recreational and cultural events; 55%; *n* = 150), welfare services 1 (various forms of occupational therapy, music therapy, art therapy, memory training; 55%; *n* = 148). It was also shown that the majority of the participants assessed the level of services provided as very good.

Other services, including, inter alia, computer classes and foreign language learning were assessed as the worst ones (very bad and bad, 3.7%; *n* = 10).

Additionally, 36% of the respondents (*n* = 99) paid attention to the lack of the remaining services, i.e., care services (using the shower, washing machine, aortic blood pressure testing, body mass measurement) and 33% (*n* = 89) of them respectively stated the lack of this kind of support.

#### 3.2.2. Assessment of Provided Services among Participants Using Day Care Centres in Different Polish Cities

In Wroclaw, participants assessed supportive services (69%; *n* = 108) and welfare services 1 (67%; *n* = 105) the highest. Similar answers were given by the respondents from Bytom who assessed the welfare services 1 (41%; *n* = 25), supportive services (40%; *n* = 24) and welfare services 2 (help in organising personal matters and doing administrative errands (38%; *n* = 23) the highest. In comparison, people from Łódź ranked meals (74%; *n* = 40) the highest.

#### 3.2.3. Significance of Provided Services

The respondents were asked to indicate three types of services that they assess as the most significant ones in the offer of day care centres, even if they are currently not provided in their facility. Almost half of the respondents (46%; *n* = 125) ranked welfare services first, second—supportive services (24% of the respondents, *n* = 66), and third—preventive services (20%, *n* = 55). Other services (activities) regarded by the respondents as significant were rehabilitation, occupational therapy, integrative activities (i.e., discussions, social games) and entertainment events. Many respondents did not fully answer the question, of whom 44 people (16%) did not give any answer. 35% (*n* = 95) of the respondents did not provide a second item, and in 49% (*n* = 132) the third answer was missing (Figure 2).

#### 3.2.4. Attendees’ Needs

The most significant services reported by study participants were the possibility to use a shower (22%; *n* = 61), obtaining nursing care (16%; *n* = 45), blood pressure, glucose level testing and body mass check (17%; *n* = 46). Cognitive training was indicated as a missing service by 14% (*n* = 39) of the respondents (Figure 3).

In this study the responses from particular cities were analysed separately, as day care centres are led by local municipalities offices, and there are no official guidelines on providing services. In Wroclaw, in contrast to other cities, the attendees more frequently (16%; *n* = 26) reported lack of care services. The respondents from day care centres in Bytom indicated a lack of activities such as cognitive training (15%; *n* = 9). The respondents from Łódź were provided with meals to the greatest extent *n* = 0 (Table 2).

#### 3.2.5. Limitations in Attending Day Care Centres

The majority of the respondents (55%; *n* = 148) did not report any significant limitations that could affect their attendance at day care centres. Some people indicated poor health condition (14%, *n* = 39), architectonic barriers—physical features that make a building inaccessible (or limit its accessibility) for people with disabilities—(9.6%, *n* = 26), and finances (9.2%, *n* = 25) as limitations to attendance.

For 20% of the attendees, the incurred costs turned out to be too high. Only 5% (*n* = 15) of the respondents did not answer the question regarding the financing of their participation at a day care centre. When comparing particular cities, there were no people exempt from the charges for attending day care centres in Łódź.

#### 3.2.6. Motivation to Attend Day Care Centres

Almost half of the respondents indicated the need for the company of other people as a reason for attending day care centres in Poland (49%; *n* = 132). Then, the respondents mentioned meals (18%), offered activities (18%) and favourable atmosphere (16%).

The comparison of the obtained data from Wrocław, Bytom and Łódź shows that the need for company is the most important reason for attending day care centres in all cities (54%—Łódź; 51%—Wrocław; 40%—Bytom).

#### 3.2.7. Potential Met Needs from the Perspective of the Attendees of Day Care Centres

The participants were asked to answer the question regarding the individual needs that should be met within the day care centres generally understood as the needs of health, care, support, psychological needs, but also social needs. Thirteen per cent of the respondents suggested that the existing offer of day care centres met their needs. Nine per cent of the respondents stated that day care centres should meet the needs for care and support (including psychological support), 6% mentioned the need for relaxation and pleasure, and 4% indicated the need to receive a meal. This question was not answered by 66% (*n* = 178) respondents (Figure 4).

The detailed analysis of the selected needs indicates that 15% of the respondents from day care centres in Wrocław, 12% of the respondents from Bytom and 6% of them from Łódź considered their needs to be met. Eighteen per cent (*n* = 11) of the respondents from Bytom believe that the need for support and care should be met at day care centres, whereas in Łódź and Wrocław this need was indicated by 7% and 6%, respectively. The attendees from Bytom also pointed out the role of meeting health needs, including facilitation of medical visits (i.e., GP)—13% respondents; provision of preventive (12%) and care services (7%). In comparison, there were no such answers among the respondents from Wrocław and Łódź. People from Łódź most frequently mentioned a guaranteed meal (13%) and meeting the need for relaxation and pleasure related to the activities organised at day care centres (9%). The latter one was also mentioned as significant by 8% of the respondents from Bytom and 5% from Wrocław.

#### 3.2.8. Open Questions

Among the whole studied group, the last question “Do you have any other comments on the functioning of a day care centre?” was answered by 42 people. Twenty four of them expressed their satisfaction with the current offer of the day care centre. Nine people indicated the need for change within the scope of the provided activities and they were illustrated with quotes, i.e., “access to the Internet, help in using new technologies”; “access to interesting books, new music CDs”; “cinema and theatre tickets”; “organisation of coach trips”; “foreign language learning, meetings with a psychologist, as well as a larger library, more frequent and diverse music therapy classes and return to dance, learning how to play chess with a professional teacher”; “access to exercise bikes, exercise mattresses.”

Seven people provided comments on the practical aspect of the functioning of day care centres and unmet needs, such as “not enough places to sit, majority of people have their own fixed places and there are not enough of them for newcomers”; “there is a need for transportation of persons who are not at full mobility”; “there is insufficient help in getting access to a day care centre”. Additionally, two people indicated the need for change within the scope of available care services, i.e., “shower access”; “lack of a nurse”.

## 4. Discussion

To the best of the authors’ knowledge, this was the first study that have critically examined the complex needs of elderly people attending day care centres in different regions in Poland. The results indicate that provided services were aimed at lonely, widowed older people (usually pensioners), who completed secondary education and had been attending day care centres for more than 5 years.

A recent systematic review from Korea was conducted to identify what types of health interventions are effective and feasible for day care centre older participants [21]. Of the 907 screened articles, 22 studies were selected. The review revealed that 59.1% of the interventions were provided by nurses, and such health interventions resulted in positive effects on senior day care centre participants’ knowledge, health behaviours, clinical indices, and hospitalization rates, but few studies within the review reported on feasibility outcomes such as needs and satisfaction [21].

In this work, the participants assessed best the meals, supportive (including social, integrative and cultural ones) and welfare services (including occupational therapy and memory training) provided. The worst assessed were additional activities like computer and foreign languages classes. On average, the respondents indicated welfare services as the most important ones. The main unmet needs at day care centres were the opportunity to take a shower and get blood pressure or blood glucose level tested. Unfortunately, only 9.91% of Polish day care centres ensure access to sanitary appliances (e.g., shower, washing machine) which would allow people to maintain personal hygiene. Slightly more than 6% of them offer blood pressure, body mass or glucose level measurement; and less than 4% ensures continuous care of a registered nurse [20].

The Department of Social and Family Affair in Poland Report [20] suggests that the level of offered services in day care centres differs based on the financial capabilities of local councils. Unfortunately, in some places, the offer was limited to a meeting place for older people and providing them with a meal. It suggests that Polish councils do not address real unmet needs of older people and the social welfare system does not efficiently cater for the growing number of older people. Moreover, only 2.36% of day care centres in Poland provides support groups meetings, moderated by a professional staff member (i.e., psychologist), dedicated to attendee’s families providing them with an opportunity to share personal experiences, feelings and as a result decrease the caregiver’s burden [20].

Our study suggests that the main reason for attending day care centres was the contact with other people. Lunt et al. [18] underline, that for older people, regular structured activities within a community setting, meeting others are thought to improve well-being and quality of life. In this work, as many as 66% of people attending day care centres assessed their needs as met. Therefore, it seems that the offered services should focus on social isolation and sense of loneliness prevention, and maintenance of social activity. These results are in line with many research results conducted among older people [22,23,24]. For many older people, the sense of isolation is huge and opportunities to engage with the wider community are limited. They are less engaged with the community. The day care centres in Poland should enable them to participate, interact with peers and have a sense of contribution to the community.

Based on the Department of Social and Family Affairs, there were 302 day care centres with 15,974 places. The same report states that the actual number of users was significantly higher—21,445 [20]. These numbers may partially explain the differences in the quality of care provided in different facilities. It also proves a great unmet need for more day care centres in Poland.

The employment structure in day care centres may also impact the quality of care. Based on the report prepared by the Supreme Chamber of Control in Poland [20], there is a visible domination of administrative and service staff members over therapeutic and care personnel. This suggests that the personnel are not qualified to provide services and work with end-users. The differences between the employment structures are also visible between particular day care centres all over Poland.

Moreover, comparing the results from different day care centres is difficult as there are no unified guidelines for their provision. A lack of coherent standards of services also impacts the possibility to assess these facilities.

This study adds to the existing knowledge by investigating user’s needs and services provided in day care centres in Poland. This analysis referred to the current lack of legislation and detailed recommendations for these facilities [20]. The presented results suggest that there is a great need for improvement in their quality in line with the users’ needs. Well-tailored care and support to the users’ needs may delay their placement in long-term care facilities, as was proven in Droes et al.’s [23] study on the Meeting Centers Support Programme. Supporting community-dwelling older people via day care activities and support for a longer time is also cost-effective [16].

The growing population of elderly people and their dependency ratio in Poland should force policymakers to better integrate currently available support services to their users’ needs. Informal home-based care, which is still the most common form of care in Poland, may soon become insufficient to provide optimal support for older adults. There is a need to find a balance between home-based care and in-patient care, using better integration of available services and strengthening support for informal caregivers [16,19,25].

There were some limitations in this study. As this was a cross-sectional study in the process of quality improvement process, we have not assessed causality. The authors of presented work have also obtained answers from day care centres located in three regions of Poland. There would be a need to reach to more than three regions to obtain a more complex picture of day care centres in Poland.

## 5. Conclusions

There is a great need for the Polish government to develop unified national standards for day care facilities, especially those dedicated to older adults. As the population ages, the policymakers in Poland must carefully consider how day DCCs can best serve each user and their caregivers with their unique circumstances, based on the holistic approach to their needs, especially the unmet ones. This step would allow improvement in their quality and close monitoring of offered services.

Robust research with the collection of meaningful outcomes is required to ensure that in Poland the increasing number of older people is enabled to access high-quality day care provision. The authors are planning another study on this topic to assess the point of view of carers of people participating in day care centres in Poland.

## Figures and Tables

**Figure 1 healthcare-08-00310-f001:**
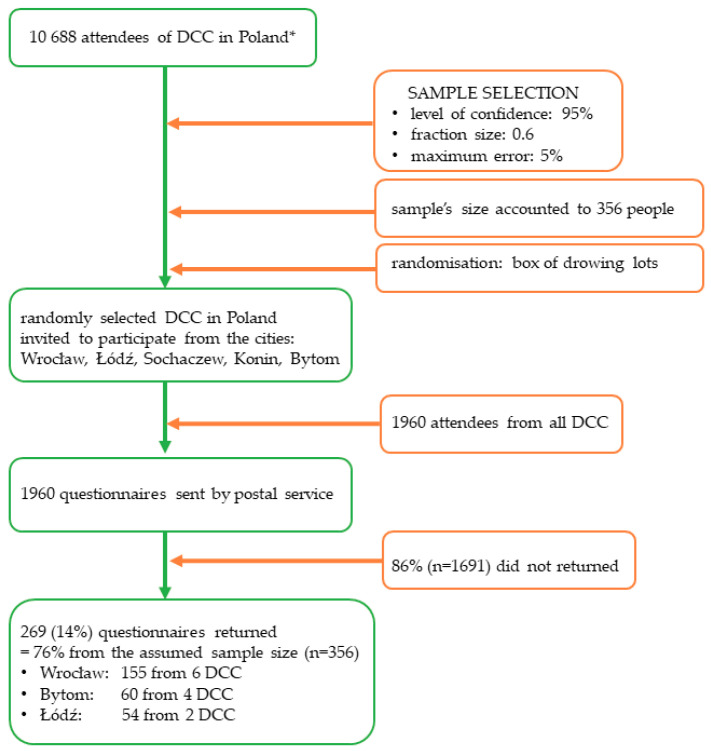
Recruitment of participants. Abbreviations: *n*, number of people; *DCC*, day care centres; %, percent. * based on the Report of the Supreme Chamber of Control in Poland, 2017 [20].

**Figure 2 healthcare-08-00310-f002:**
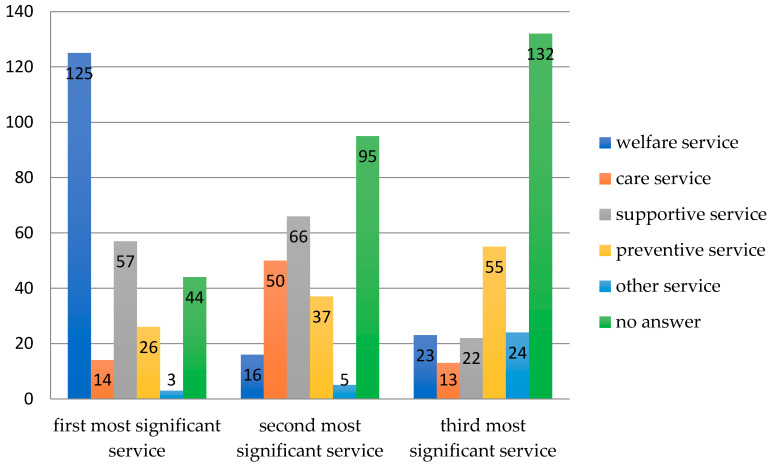
Assessment of the significance of services in the opinion of attendees of day care centres in Poland.

**Figure 3 healthcare-08-00310-f003:**
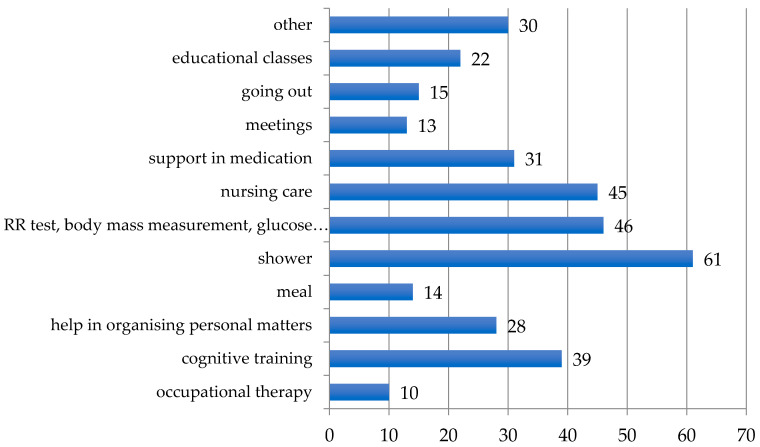
Unavailable services in the opinion of the attendees of day care centres in Poland. Abbreviations: RR, blood pressure.

**Figure 4 healthcare-08-00310-f004:**
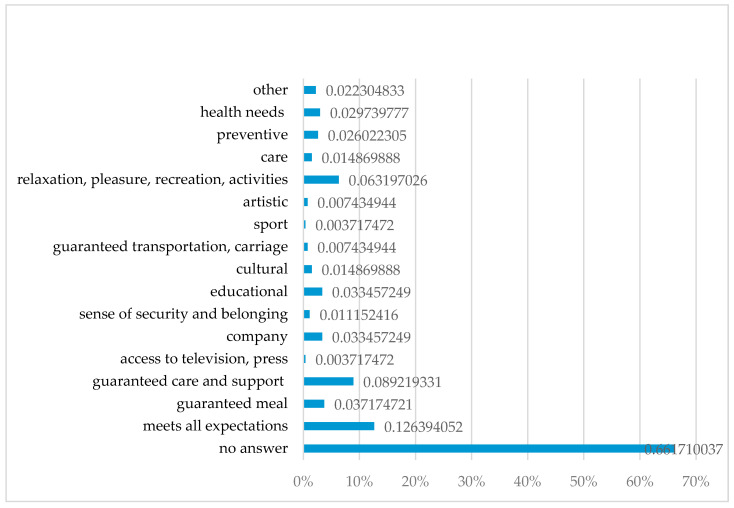
Potential needs met at day care centres in Poland from the perspective of the attendees. Abbreviations: %, percent.

**Table 1 healthcare-08-00310-t001:** Sociodemographic data of the attendees of day care centres in Poland.

Feature	All Respondents (*n* = 269)	
age	74 (± 11)	
	No answer *n* = 24 (9%)	
sex	Women	164 (61%)
	Men	98 (36%)
	No answer	7 (3%)
education	Primary	59 (22%)
	Vocational	81 (30%)
	Upper secondary	100 (37%)
	Higher	15 (6%)
	No answer	14 (5%)
marital status	Single	52 (19%)
	Married	29 (11%)
	Widow/widower	136 (51%)
	Divorced	32 (12%)
	Informal relationship	6 (2%)
	Other	4 (1%)
	No answer	10 (4%)
previously practised profession	White-collar worker	63 (23%)
	Blue-collar worker	86 (32%)
	Worker performing both manual and non-manual work	33 (12%)
	Never worked	9 (3%)
	No answer	77 (29%)
obtained type of pension	Pension	176 (65%)
	Disability pension	60 (22%)
	Other	20 (7%)
	No answer	13 (5%)
place of residence	City above 300 thousand	181 (67%)
	City between 100–300 thousand	43 (16%)
	City below 100 thousand	27 (10%)
	Village	1 (0.4%)
	No answer	17 (6%)
time of using day care centre services	More than 5 years	129 (48%)
	More than 2 years	59 (22%)
	1–2 years	38 (14%)
	Half a year—year	17 (6%)
	Less than half a year	16 (6%)
	No answer	10 (4%)
availability of day care centres ^1^	Very good	99 (37%)
	Good	76 (28%)
	Mediocre	41 (15%)
	Bad	14 (5%)
	Very bad	2 (1%)
	No answer	32 (14%)

Abbreviations: *n*, number of people; ±, standard deviation; %, percent. (^1^) refers to how respondents assessed the availability of day care centres in their city.

**Table 2 healthcare-08-00310-t002:** Unavailable services in the opinion of the attendees of day care centres in Poland divided according to cities: Łódź, Bytom, Wrocław.

Missing Service	Wrocław	Bytom	Łódź
Occupational therapy	5 (3%)	3 (1.8%)	2 (3.7%)
Memory trainings	13 (8%)	16 (26.6%)	10 (18.5%)
Help in organising personal matters	13 (8%)	8 (13%)	6 (11%)
Meal	9 (5.8%)	5 (8%)	0
Shower	29 (18.7%)	15 (25%)	17 (31%)
RR test, body mass measurement, glucose testing	22 (14%)	12 (20%)	6 (11%)
Care	26 (16.7%)	8 (13%)	8 (14.8%)
Support in medication	16 (10%)	6 (10%)	5 (9%)
Meetings	3 (1.9%)	7 (11%)	3 (5.5%)
Going out	3 (1.9%)	5 (8%)	2 (3.7%)
Educational classes	6 (3.8%)	9 (15%)	1 (1.8%)
Other	11 (7%)	6 (10%)	0

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
