# Peer review of "Needs of Older People Attending Day Care Centres in Poland"

_healthcare, 2020, doi:10.3390/healthcare8030310_

Round 1
Reviewer 1 Report
This article analyzes the requirements of older people attending day care centers in Poland. The article is in the scope of the journal, but redaction and structure should be improved as indicated below. Especially the methods and results should be clearer; the author is recommended to identify and practice sophisticated objectives for a journal publication. It was impossible for me to identify the novelty or even the actual scientific contribution of the paper. The author must justify the following points:
Comment 1: A concise and factual abstract is required where the author needs to answer the following questions: What is the aim of this work? What methods did you use? What are the main results? And what conclusions can you draw from your results? The paper should be revised to highlight novelties. Please consider that this lack of novelty starts with the Abstract, Introduction, and Conclusion. The aim of this work should be clarified more clearly. What is the importance of this paper? The introduction must include motivation and background. Besides, a literature review of recent scientific papers covering only the topic and leading to the submission hypothesis based on the gap analysis of the previously published research is required.
Comment 2: In line 42 it says “Particularly, these actions should facilitate social activation, including …” which actions? please specify. Lots of information in the introduction needs to be cited and explained.
Comment 3: I would strongly suggest adding a figure at the beginning of Section 2 to describe the processed methods and applied materials in this part of the study in a way to facilitate the understanding process for readers.
Comment 4: As the questionnaire is the main source of data collection for this work, I would suggest adding it as a supplementary document where the reader could make an access and prove the results and discussion.
Comment 5: The materials and methods section is not outlined with necessary vigor, and the subsections need to be rewritten and relocated within the manuscript. The author needs to include sufficient methodological details in the paper and elaborate on the produced results from the proposed methodology. The paper should provide enough information to be replicable by other researchers. Further, in line 124 it says “The survey was developed based on the official care guidance for day care centres in Poland” The author must briefly illustrate the official care guidance for day care centers in Poland. Besides, all subsections presented in Section 3 must be identified and well presented in Section2 before being evaluated in Section3.
Comment 6: Section 3 must be entitled “Results and Discussion” while the last section is “Conclusions”. In this case, several paragraphs must be rewritten and reorganized where Section 3 should be improved by including a clear and concise analysis of all results presented. Further, the author must justify the selection of features presented in Table 1. How the significance of services has been evaluated and classified in Figure 1? What has this work done to meet the requirements of day care centers in Poland?
Comment 7: The Conclusion section should highlight the novelty and the materials and methods used in this work and point out the collected results. Then, present a summary of the limitations of this research as well as the recommendation for future works.
Comment 8: The author is using (we) too much. Please consider that this is a scientific journal publication, where you need to avoid some phrases like (we, our, ….). Instead, you can use (this work, this study, this analysis….). Besides, proofreading by a native English speaker should be conducted to improve clarity and organization quality.
Reviewer 2 Report
Please read all my comments in each part of the manuscript in the form of pdf notes.

Reviewer 3 Report
1.- Exclusion Criteria?
What about of persons with Dementia or MCI, may be they can´t answer
correctly.
2.- I would like to see the survey. How long the persons used in order to answer?
3.- Study Design
First you say it´s a Quality improvement process, but later you say it is a Cross
Sectional Study. Please define.
4.- In relation to your objective.
At final, you have answers of older persons from 3 Day Care Centres.
Do you think, it could be better to evaluate centre by centre independently?
Because the number of participants were different too: Wroclaw: 155 responders, Lodz :54 and Bytom: 60
And when you finish the evaluation of each centre, you would do a final comparison.
5.- A question. Do The Day Care Centres have the obligation to give a good service, to have good structures, good processes and a determined capacity for the users?
Is there a regulation? is there a supervision by the government or by the municipality?
Is it important to know the results of supervision of these authorities, before to do this research?.
6.- Is it important also to know the opinion of the relatives of Older Persons about these Day Care Centres?
7.- What is the real number of questions that Older Persons didn't answer in each of these 3 centres?
8.- What was the instrument that you used in order to say the Older Person receive good attention or not?
9.- The survey was self-applied? What happened if the older persons didn't understand the question? Somebody did explain them the idea? or not?
10.- In the results and discussion, you say that these centres need to improve the processes in order to give better attention.
But in the abstract you say, that the answer of the responders was: Very Good and Good.
So there is a contradiction. Please explain me.
Reviewer 4 Report
- A representative of the analyzed data should be discussed. Do daycare center attendees include "demented" ones? if so, I am afraid that the self-administered questionnaire is not an appropriate method to collect their needs.
- In the introduction, please explain the reason why you compared different cities.
- Please modify "memory training" in Figure2 to "cognitive training"
- It is difficult to understand what is different between "... significant services reported by study participants (line 190)" and "Participants were asked to answer the question regarding the needs that should be met within the daycare centres (line 218)". I can understand the former is about "service needs based on current service content" and the latter is about "subjective needs not based on current service content". If so, please explain as readers can understand it easier and clearer.
Round 2
Reviewer 1 Report
The work has really developed and the author answered most of my previous comments. However, the author still needs to reconsider the comment (6) in my previous review, where the Discussion must be separated than the Conclusion. It could be inserted as an individual section or be emerged with the Result section. In this case several paragrahs must be relocated.
Author Response
REPLY TO COMMENTS OF THE REVIEWER 1
Reviewer #1:
The work has really developed and the author answered most of my previous comments. However, the author still needs to reconsider the comment (6) in my previous review, where the Discussion must be separated than the Conclusion. It could be inserted as an individual section or be emerged with the Result section. In this case several paragrahs must be relocated.
RESPONSE: Thank You very much fot this suggestion. Of course we agree with this remark and we decided to separate the Discussion and Conclusion parts (they are inserted as individual sections).
We are very glad, that our manuscript developed. Thank You.
Authors.